# The Omicron Variant Is Associated with a Reduced Risk of the Post COVID-19 Condition and Its Main Phenotypes Compared to the Wild-Type Virus: Results from the EuCARE-POSTCOVID-19 Study

**DOI:** 10.3390/v16091500

**Published:** 2024-09-23

**Authors:** Francesca Bai, Andrea Santoro, Pontus Hedberg, Alessandro Tavelli, Sara De Benedittis, Júlia Fonseca de Morais Caporali, Carolina Coimbra Marinho, Arnaldo Santos Leite, Maria Mercedes Santoro, Francesca Ceccherini Silberstein, Marco Iannetta, Dovilé Juozapaité, Edita Strumiliene, André Almeida, Cristina Toscano, Jesús Arturo Ruiz-Quiñones, Chiara Mommo, Iuri Fanti, Francesca Incardona, Alessandro Cozzi-Lepri, Giulia Marchetti

**Affiliations:** 1Clinic of Infectious Diseases, San Paolo Hospital, ASST Santi Paolo e Carlo, Department of Health Science, University of Milan, 20142 Milan, Italy; francy.bai19@gmail.com (F.B.); andrea.santoro1@unimi.it (A.S.); 2Division of Infectious Diseases, Department of Medicine Huddinge, Karolinska Institute, 17177 Stockholm, Sweden; pontus.hedberg@ki.se; 3Icona Foundation, 20145 Milan, Italy; alessandro.tavelli@icona.org (A.T.); sara.debenedittis@icona.org (S.D.B.); 4School of Medicine, Federal University of Minas Gerais, Belo Horizonte 30130-100, Minas Gerais, Brazil; julietcaporali@hotmail.com (J.F.d.M.C.); carolinacmarinho@gmail.com (C.C.M.); asleite.medicina.ufmg@gmail.com (A.S.L.); 5Department of Experimental Medicine, University of Rome Tor Vergata, 00133 Rome, Italy; santormaria@gmail.com (M.M.S.); ceccherini@med.uniroma2.it (F.C.S.); marco.iannetta@uniroma2.it (M.I.); 6Vilnius Santaros Klinikos Biobank, Vilnius University Hospital Santaros Klinikos, 08406 Vilnius, Lithuania; dovile.juozapaite@santa.lt; 7Clinic of Infectious Diseases and Dermatovenerology, Institute of Clinical Medicine, Medical Faculty, Vilnius University, 03101 Vilnius, Lithuania; edita.strumiliene@santa.lt; 8Centro Universitário de Lisboa Central, Centro Clínico Académico de Lisboa, 1169-050 Lisboa, Portugal; andre.almeida@chlc.min-saude.pt; 9Centro Hospitalar de Lisboa Ocidental, 1449-005 Lisboa, Portugal; ctoscano@ulslo.min-saude.pt; 10Hospital Juan Graham Casasus, Villahermosa 86126, Tabasco, Mexico; drjesusruiz@gmail.com; 11EuResist Network GEIE, 00152 Rome, Italy; chiara.mommo@euresist.org (C.M.); iuri.fanti@gmail.com (I.F.); f.incardona@informa.pro (F.I.); 12InformaPRO S.R.L., 00152 Rome, Italy; 13Centre for Clinical Research, Epidemiology, Modelling and Evaluation (CREME), Institute for Global Health, UCL, London WC1E 6BT, UK; a.cozzi-lepri@ucl.ac.uk

**Keywords:** post COVID-19 condition, long COVID, post acute sequelae of SARS-CoV-2 infection, SARS-CoV-2 viral variant, omicron variant

## Abstract

Post COVID-19 condition (PCC) is defined as ongoing symptoms at ≥1 month after acute COVID-19. We investigated the risk of PCC in an international cohort according to viral variants. We included 7699 hospitalized patients in six centers (January 2020–June 2023); a subset of participants with ≥1 visit over the year after clinical recovery were analyzed. Variants were observed or estimated using Global Data Science Initiative (GISAID) data. Because patients returning for a post COVID-19 visit may have a higher PCC risk, and because the variant could be associated with the probability of returning, we used weighted logistic regressions. We estimated the proportion of the effect of wild-type (WT) virus vs. Omicron on PCC, which was mediated by Intensive Care Unit (ICU) admission, through a mediation analysis. In total, 1317 patients returned for a post COVID visit at a median of 2.6 (IQR 1.84–3.97) months after clinical recovery. WT was present in 69.6% of participants, followed by the Alpha (14.4%), Delta (8.9%), Gamma (3.9%) and Omicron strains (3.3%). Among patients with PCC, the most common manifestations were fatigue (51.7%), brain fog (32.7%) and respiratory symptoms (37.2%). Omicron vs. WT was associated with a reduced risk of PCC and PCC clusters; conversely, we observed a higher risk with the Delta and Alpha variants vs. WT. In total, 42% of the WT effect vs. Omicron on PCC risk appeared to be mediated by ICU admission. A reduced PCC risk was observed after Omicron infection, suggesting a possible reduction in the PCC burden over time. A non-negligible proportion of the variant effect on PCC risk seems mediated by increased disease severity during the acute disease.

## 1. Introduction

After the acute phase of COVID-19, a varying number of patients, estimated to be about 10–40%, experience persistent symptoms following viral clearance [1,2]. Since the onset of the pandemic, various definitions of long COVID have been used in the scientific literature, including long COVID and Post Acute Sequelae of SARS-CoV-2 (PASC), leading to significant variability in the reported prevalence of this condition [3,4]. To avoid further ambiguity, the World Health Organization (WHO) set out, in October 2021, to identify a standardized definition: “Post COVID-19 condition” is defined as the continuation or development of new symptoms 3 months after the initial SARS-CoV-2 infection, with these symptoms lasting for at least 2 months and no other explanation. More generally, Centers for Disease Control and Prevention (CDC) defines “Post COVID Conditions” (PCC) as the physical and mental health consequences that are present 4 or more weeks after SARS-CoV-2 infection [5,6].

A common clinical presentation of this condition draws similarities to myalgic encephalomyelitis or chronic fatigue syndrome (ME/CSF), which is itself a typical manifestation of the broader group of “post-acute infection syndromes (PAIs)” [7,8]. Moreover, symptoms can fluctuate and relapse over time, and even last years after the acute phase [6,9]. In order to promptly identify patients who could suffer from persistent complications of COVID-19, a more precise definition and the identification of clusters of symptoms could prove useful.

With regard to possible predictors, PCC seems to have higher chances of occurring after a severe acute disease, but cases following mild to moderate illness have also been reported [10]. Initial data reported a possible protective role of vaccination and antiviral treatments in reducing the risk of PCC [11,12,13,14]: previous works have demonstrated a reduction in PCC after vaccination between 15% and 41%, and evidence suggests that treatment with nirmatrelvir/ritonavir during the acute phase may reduce PCC incidence [15,16,17,18,19].

In the first phases of the pandemic, no changes or small reductions in the prevalence of PCC were observed between the ancestral strain and the Alpha variant, as reported by a previous study, which compared long COVID symptoms between March and December 2020 (original Wuhan strain) and January to April 2021 (prevalent Alpha variant) [20], but reductions in the risk of PCC emerged with the most recent variants, mainly in comparison with the original wild-type strain of the virus: a decrease in the incidence of PCC among individuals who contracted the Delta and Omicron variants in contrast to those infected with the wild-type virus has been described [20,21,22,23,24,25,26]. Some authors also demonstrated a reduced risk of developing PCC after Omicron infection (December 2021–March 2022) compared to the Delta variant (June 2021–November 2021), with an overall odds reduction ranging from 0.24 to 0.50, depending on age and time since vaccination. A modification of PCC clusters over time has also been observed, although not in all previous studies: in fact, fatigue remains the most common symptom in PCC, followed by neurological symptoms, while long-term cardiorespiratory symptoms and anosmia/dysgeusia seem to be less frequent after the Omicron infection [23,27,28,29].

No specific treatment is currently available for PCC or ME/CSF, placing a heavy burden on already stressed health systems; furthermore, a recent study has shown that post COVID-19 patients had higher mortality rates, with an excess death rate of 16.4 per 1000 individuals [30].

The European Cohorts of Patients and Schools to Advance Responds to Epidemics (EUCARE) is a multicenter study investigating the clinical, epidemiological, virological and immunological aspects of COVID-19 epidemics through the collaboration of a comprehensive multidisciplinary team; the EuCARE-POSTCOVID study focuses on investigating the possible predictors of PCC and the role played by viral variants in the development of this condition.

In this scenario, additional analyses are needed to evaluate whether specific SARS-CoV-2 viral strains may play a role in the development of PCC or specific symptom clusters, and also to evaluate whether severe disease during the acute phase and the consequent admission to an Intensive Care Unit (ICU) is a key event in the pathway from viral infection to the development of PCC.

## 2. Materials and Methods

### 2.1. Study Design and Study Population

The EuCARE POSTCOVID study is a retrospective and prospective multicenter cohort study nested within the EuCARE hospitalized cohort; the EuCARE hospitalized and the EuCARE POSTCOVID cohort have been described in detail elsewhere [31,32]. All participants of the hospitalized cohort who were discharged from hospital after acute SARS-CoV-2 infection were offered to come back for a post COVID visit, which was performed in the year after clinical recovery: at 2–3 months (entry in the POSTCOVID cohort, T0), 6–9 months (T1) or 12–15 months (T2).

The study flow chart is illustrated in Appendix A. We included in this analysis hospitalized patients with a confirmed diagnosis of SARS-CoV-2 infection from 4 January 2020 to 07 June 2023 in 6 centers over 3 continents enrolled in the EuCARE hospitalized cohort. After clinical recovery, a subset of these patients returned for at least one visit at a post COVID-19 outpatient clinic in each center, or were followed-up by telemedicine, and were thus included in the EuCARE POSTCOVID cohort. At the time of this analysis, not many participants have been followed-up with past an entry visit, so the analysis is essentially cross-sectional at T0. The participating centers are as follows: Clinic of Infectious Diseases, San Paolo Hospital, ASST Santi Paolo and Carlo, Department of Health Sciences, University of Milan, Italy; Vilnius University Hospital, Santaros Klinikos, Vilnius, Lithuania; Policlinico Tor Vergata, Università degli Studi di Roma Tor Vergata, Rome, Italy; Regional Hospital Dr. Juan Graham Casasús, Villahermosa, Tabasco, Mexico; and Centro Hospitalar de Lisboa Ocidental, Lisbon, Portugal and Federal University of Minas Gerais, Minas Gerais, Brazil. We excluded patients who died during hospitalization and who refused to be followed-up with into the POSTCOVID cohort.

### 2.2. Primary and Secondary Objectives

The main hypothesis of this study was that the risk of PCC might be reduced following infection with recent viral variants, compared to that seen in the early waves of the pandemic. The experience of a lower severity of COVID-19 during the acute phase is one of the possible reasons that could explain this reduction in risk over time, which we have also investigated.

Thus, the primary objective of our analysis was to evaluate the association between SARS-CoV-2 variants at time of primary SARS-CoV-2 infection and the risk of developing PCC after recovering from acute infection.

The secondary aims were as follows: (i) to investigate whether the predictive role of the variants might be different according to specific PCC phenotypes/clusters (i.e., brain fog, respiratory symptoms and fatigue), and (ii) to estimate the proportion of the total effect of variants on the risk of developing PCC, which could be explained by disease severity during the acute phase.

### 2.3. Study Procedures

After the acute phase, patients underwent routine blood exams (whole blood count, creatinine, SGOT, SGPT, C reactive protein—CRP) and a comprehensive medical visit. All patients filled in a short version of the post COVID-19 WHO Case Report Form to record their symptoms under a physician’s supervision [33]. We asked if the patients experienced any symptoms, after the acute disease and their discharge from the hospital, that were not experienced prior acute SARS-CoV-2 infection; for each symptom, we recorded if it had already been resolved, if it was still ongoing or if it was present only sometimes.

### 2.4. Definitions and Data Collection

The primary outcome was a proportion of participants being diagnosed with PCC, as per the CDC definition: the presence of at least one new or persistent symptoms ≥1 month after the acute infection. Differential diagnoses that might explain these persistent symptoms were excluded based on the physician’s opinion. We focused on the main CDC definition as well as the secondary outcomes encompassing 3 proposed clusters of symptoms [34,35,36]: (i) fatigue; (ii) respiratory symptoms and (iii) brain fog/central nervous symptoms (Appendix A). All data on the acute phase and the follow-up visit at the post COVID-19 clinic were collected using a standardized electronic Case Report Form (CRF).

Concerning viral variants, viral sequences were available for a subset of patients (the whole genome has been sequenced, often including multiple genomic regions or only the spike gene, according to the methodology adopted in the different participating centers during the study period). When a viral sequence was not available in the EuCARE database, the variant was inferred using the variant prevalence data reported in the publicly available Global Data Science Initiative (GISAID) dataset [37]; in particular, we assigned the variant that was most frequently circulating in the participants’ geographical region using a moving 1-week time window around the date of SARS-CoV-2 infection or hospitalization, whichever was available. We only included participants who were assigned wild-type SARS-CoV-2 or one of the following viral variants: Alpha, Gamma, Delta, or Omicron.

### 2.5. Statistical Analyses

Categorical data are presented as absolute numbers and percentages, quantitative variables as medians and interquartile ranges (IQRs). We compared characteristics across groups (participants with and without follow-up visits and according to the assigned viral variants) using the Chi-square, Mann–Whitney or Kruskal–Wallis test, as appropriate.

The proportion of participants who were diagnosed with PCC was calculated according to variants, and a logistic regression model was used to estimate the odds ratio (OR) of developing PCC, using the wild-type strain as the comparator. We fitted 4 sets of separate models, one for each of the pre-specified outcomes: the CDC’s main definition of PCC, fatigue, respiratory sequelae and brain fog clusters.

We identified the following confounders, which have been included in the multivariable logistic regression models: age, sex, pre-existing comorbidities (as binary variables: at least 1 comorbidity vs. no comorbidities) and time of infection. In the analysis exploring the risk of PCC according to viral variant, the calendar time of infection was also modeled with restricted cubic splines, using three knots: 1 July 2020, for the introduction of Dexamethasone in the therapy of COVID-19, thanks to the RECOVERY trial results [38], November 1st 2020 for the Alpha variant circulation and 30 April 2021 for the Delta variant circulation. Vaccination (defined as having received at least 2 doses of an mRNA vaccine) is a possible confounder and effect measure modifier for the association of interest, as participants who underwent vaccination were more likely infected with variants other than the wild-type virus have a lower risk of severe disease, and might be protected against PCC. We addressed this issue by performing a sensitivity analysis restricted to the unvaccinated population. The full set of confounders was identified using the results from randomized studies and previous axiomatic knowledge, and included all measured common causes of exposure and outcomes [12,39].

Because it is conceivable that symptomatic patients of the EuCARE hospitalized cohort were more likely to return to the clinics for a post COVID-19 evaluation visit, by restricting the analysis to this subgroup, collider bias might have been introduced by sampling; indeed, it is possible that the exposure of interest (SARS-CoV-2 viral variant) could also modify the probability of returning for a post COVID-19 evaluation, either via greater severity of disease or because lockdown measures changed over different waves of the pandemic. The directed acyclic graph exploring this possible collider bias is shown in Appendix A. Because the study population of the POSTCOVID cohort was nested within the data environment of all participants in the hospitalized cohort, we were able to use inverse probability weighting to try to minimize this bias. Specifically, we created a pseudo-study population in which participants were weighted according to the inverse of their probability of being sampled for a post COVID-19 visit. The propensity score model used to generate the weights included the key predictors of COVID-19 outcomes: sex, age, comorbidity and variant. Since a pro-inflammatory status in the acute phase has been associated with disease severity and COVID-19 outcome [4], for a subgroup for which a measure of blood C reactive protein (CRP) at hospitalization was available, we created an alternative set of weights after including this marker in the propensity score models.

Finally, we aimed to evaluate whether the effect of viral variant on the risk of PCC could be explained by the severity of COVID-19 disease during the acute phase. We thus conducted a 4-way decomposition method for the mediation analysis to estimate what proportion of the total effect associated with the wild-type virus (using the Omicron variant as a comparator) on PCC might be mediated by disease severity. The need for mechanical ventilation (MV) and Intensive Care Unit (ICU) admission vs. no oxygen therapy and the use of Continuous Positive Airway Pressure (CPAP), or Non-Invasive Mechanical Ventilation (NIV) vs. no oxygen treatment (in 2 separate models), during the acute phase was used to classify participants with severe and moderate, respectively, and used as a mediator for the difference in PCC risk between the Wuhan and Omicron strains. Statistical analyses were performed using SAS software (version 9.4, Carey, NC, USA).

## 3. Results

### 3.1. EuCARE Hospitalized Cohort

A total of 7699 patients were hospitalized for SARS-CoV-2 infection from 4 January 2020, to 7 June 2023, and subsequently discharged (EuCARE hospitalized cohort, Appendix A). Most of the 7699 participants exhibited the wild-type virus (3014 participants, 39.2%), followed by the Omicron (29.9%), Alpha (16.6%) and Delta variants (12.7%); lastly, only a minority of participants were infected with the Gamma variant (1.6%). Overall, the participants had a median age of 64 years (IQR 51–77), with 3358 (43.6%) being female. Almost one-third of the participants were Italian (2437, 31.7%). The majority of participants (5006, 65%) presented with ≥1 pre-existing comorbidity, with diabetes, hypertension, cardiovascular and kidney diseases, and cancer being the most commonly reported. Notably, only a minority of participants (878, 11.4%) had received two or more doses of the COVID-19 vaccine, as the majority became infected and were enrolled in this study before vaccination was widely available.

### 3.2. EuCARE POSTCOVID Cohort Nested in the Hospitalized Cohort

Among the 7699 patients of the EuCARE hospitalized cohort, 1317 (17.1%) underwent at least one post COVID-19 evaluation over the year after the acute phase, thus qualifying for inclusion in the EuCARE-POSTCOVID cohort (Appendix A). The median time for follow-up visits after hospital discharge and/or clinical recovery was 2.6 months (IQR 1.84–3.97).

To assess the potential impact of collider bias on our analysis, we compared hospitalized patients who attended ≥1 follow-up visit and entered the POSTCOVID cohort with those who never returned for evaluation: the former group tended to be younger and with a lower prevalence of pre-existing comorbidities, although asthma, obesity and smoking were more common in PCC cohort participants compared to patients who never returned for a PCC visit (Table 1).

Finally, participants in the POSTCOVID cohort exhibited lower vaccination rates and higher levels of inflammation, as indicated by elevated CRP levels during the acute phase.

Regarding viral variants, 37/1317 (2.8%) variants were classified by viral sequencing, while, for the remaining participants, variants were estimated with the GISAID repository [37]. The predominant viral variant among patients included in the POSTCOVID cohort was the wild-type virus (69.6%), followed by Alpha (14.4%) and Delta (8.9%), while Omicron was shown in 3.3% (Table 1). The Omicron subvariants detected were BA.1 in 8 (19%), BA.2 in 15 (35%), BA.5 in 15 (35%) and BQ.1.1 in 5 (11%) patients.

Table 2 shows the characteristics of participants of the POSTCOVID cohort, according to the viral variant.

Participants infected by the Omicron variant were generally older, females, Italian and more likely to have at least one pre-existing comorbidity, with the exception of diabetes, heart diseases, obesity and smoking, that was more common in participants infected by the wild-type variant. Furthermore, the Omicron variant was associated with higher vaccination rates, as expected, and lower inflammation at hospital entry, compared to the wild-type strain (Table 2).

### 3.3. Risk of PCC According to Viral Variant from Fitting a Logistic Regression Model

We found that 918/1317 (69.7%) participants met the criteria for PCC. Most PCC diagnoses (854/918, 93%) occurred at 2–3 months after the acute phase (T0). Regarding clusters of symptoms, fatigue (638/1317, 48.4%) was the most common PCC presentation, while one-third of participants complained of brain fog (401/1317, 30.5%) or respiratory symptoms (514/1317, 39%).

Participants diagnosed with PCC were more likely to be female (436, 47.5% vs. 130, 32.6%, *p* < 0.001) and presented more frequently with pre-existing comorbidities (590, 64.3% vs. 209, 52.4%, *p* < 0.001), compared to participants without PCC. When compared to the original wild-type strain, a lower proportion of PCC was observed with the Omicron variant in this unadjusted analysis (Table 3).

Table 3 shows the results of the univariable and multivariable analysis using a logistic regression model.

After adjustments for age, gender, comorbidities and calendar time (using restricted cubic splines), the Omicron variant was associated with a reduced risk of PCC compared to the wild-type virus, even in the unweighted model (not controlling for collider bias). Conversely, we observed a higher risk of PCC with the Alpha and Delta variants compared to the wild-type virus: the Alpha variant was associated with a higher PCC risk in both the unweighted analysis, which controlled for confounding bias alone, and in the weighted model, which attempts to also minimize collider bias, while the contrast for the Delta variant vs. wild type showed a trend towards statistical significance in the weighted model. Infection with the Gamma variant also carried a higher risk of PCC in all models (Table 3). After restricting the analyses to a subset of the unvaccinated population, a reduced risk of PCC was confirmed following the Omicron infection, while the Alpha and Gamma variants were associated with a higher risk of PCC compared to the wild-type infection, although without statistical significance in the adjusted weighted model (Appendix A).

### 3.4. Risk of Chronic Fatigue, Brain Fog and Respiratory Sequelae According to Viral Variant from Fitting a Logistic Regression Model

When we examined the secondary clusters of PCC, the outcome results were similar to those of the main analysis (Table 4).

In particular, we observed that the Omicron variant compared to the wild-type virus was associated with a reduction in all PCC phenotypes. Furthermore, the Alpha variant was associated with around a four-fold higher risk of fatigue and a two-fold higher risk of respiratory complications and brain fog compared to the wild-type strain. The Delta variant was associated with a higher risk of fatigue in all models in comparison with the wild-type virus; a weaker association was reported between the Delta variant and respiratory symptoms or brain fog, reaching statistical significance only in the adjusted weighted model. The Gamma variant was associated with a higher risk of respiratory sequelae in the weighted models. We also used the conservative significance level of 0.0125 (Bonferroni-adjusted *p*-value 0.05/4 = 0.0125) and the data still carry evidence that, compared to participants infected with the wild-type strain, those infected with the Omicron variant were at lower risk, while those infected with the Alpha and Delta variant were at higher risk of developing PCC.

Finally, a reduced risk of brain fog and fatigue was displayed following infection with the Omicron variant, while a higher risk of brain fog was observed with the Alpha variant, a higher risk of fatigue with the Alpha and Delta variant and a higher risk of respiratory symptoms with the Gamma variant, also restricting the analyses in the unvaccinated patients (Appendix A).

### 3.5. Mediation Analysis for the Contrast Wild-Type Virus Compared to Omicron Variant and PCC

A subset of 401/959 participants (30%) infected either with the wild-type (n = 384, 96%) or the Omicron (n = 17, 4%) variant were included in the counterfactual mediation analysis evaluating ICU/MV vs. no oxygen treatment. Of these, 134 (33%) were admitted to an ICU or received MV during the acute phase, with the majority (96%) in the wild-type group. A total of 113/134 of the ICU participants (84%) developed a PCC vs. 267/196 (73%) of those who were not admitted to an ICU (*p* = 0.01). The formal mediation analysis showed that 41.7% (14.0–69.5%) of the total effect of the wild-type virus vs. Omicron variant on the risk of PCC was mediated by ICU admission and MV (vs. no oxygen therapy). However, almost a null proportion of the effect of the wild-type strain vs. Omicron variant on the PCC risk was explained by CPAP or NIV (vs. no oxygen). The same analysis was conducted using brain fog as the outcome, and, of note, we found that only a small proportion of the total effect of the variant on PCC was caused by disease severity (Table 5).

## 4. Discussion

In our multicentric cohort of hospitalized patients who had a follow-up visit after the acute disease, we found that the following:(i)The Omicron variant was associated with a reduction in the risk of PCC, while the Alpha and Delta variants were characterized by a higher risk of PCC, compared to the wild-type virus.(ii)The direction and magnitude of the associations seen for variants and the CDC PCC outcome were similar when we evaluated the risk of specific PCC clusters (brain fog, respiratory symptoms and fatigue).(iii)An important proportion of the total effect of the Omicron versus wild-type virus on the risk of PCC appeared to be explained by disease severity in the acute phase.

Other previous analyses confirmed the reduction in the risk of PCC and its main clusters of symptoms, as well as the decrease in the number of persistent symptoms, following infection with the Omicron variant compared to the wild-type strain [20,21,27,28,30,40,41,42]. The lower risk observed with the Omicron variant, compared to the wild-type virus, might explain the reduction in incidence of new cases recently observed [43]. The reduction in the risk of PCC with the Omicron compared to Alpha and Delta variants was confirmed in several previous studies with similar ORs [21,23,25,27], although not all studies found a sharp difference between the Omicron and Delta virus [42,44]. In contrast, our results are at odds with the published literature, as the Wuhan strain was previously associated with a similar or a slightly higher PCC risk compared to the Alpha variant, and a higher risk compared to Delta and subsequent variants [24,45]. The reasons for this discrepancy are unclear, but we cannot rule out selection or other sources of bias (the majority of participants infected with the Gamma variant were enrolled in Brazil), despite our attempt to control for these using a multivariable weighted analysis. Furthermore, specific phenotypes of PCC might have triggered these associations, as a reduction in the persistence of anosmia and dysgeusia vs. an increase in brain fog, myalgia and anxiety/depression symptoms over time has been described [45].

Our estimated risk of PCC is over-estimated as compared to that currently shown in the literature [46]; recent studies displayed the presence of PCC in 30% of patients at 3–6 months and at a long-term follow up of 2 years [3,47]. Possible contributions to our higher PCC incidence include the inclusion of a selected sample of participants who returned for a post COVID-19 evaluation because they were probably symptomatic; in fact, the healthier participants and those who completely recovered from COVID-19 are likely not to be included here. Furthermore, we used a broad definition of PCC that included any symptom at least 1 month after the acute disease. Finally, we included only previously hospitalized patients compared to other studies, which also included outpatients; nevertheless, we acknowledge that in the first wave of the pandemic, several hospital admissions were carried out in patients with mild symptoms as well. However, the difference in the risk of PCC according to variants is likely to be robust, as we did our best to try to minimize both confounding and collider bias by applying regression modeling inverse probability weighting.

One strategy to better classify patients with PCC and to understand the best diagnostic and therapeutic procedures was to use previously proposed clusters of symptom definitions [34]. Among the three clusters considered, the most common clinical phenotypes were fatigue, which was observed in half of participants and has several aspects in common with ME/CFS, followed by brain fog or respiratory sequelae, which were observed in one-third of the cohort, consistently with what was shown in previous works [48,49,50,51,52]. ME/CFS could follow several infections, not just SARS-CoV-2, and because several symptoms overlap between the two conditions, it has recently been recognized as one of the possible manifestations of PCC; the identification of these patients, especially of the most severe ones, and their correct management is essential, since the syndrome has a significant impact on the patients’ quality of life [53,54,55,56,57].

The exact mechanisms by which infection with a specific variant may affect the risk of developing PCC remain unclear. These include the viral pathways (more severe symptoms in the acute phase, higher risk of hospitalization), other pathways linked to stress (due to the need for treatment in an ICU, common to other infections) or to other socio-economic risk factors, such as days at work lost, lower income and an inability to adequately rest in the early weeks after developing COVID-19. Our analysis carried evidence that ICU admission appeared to explain approximately 40% of the total effect attributable to the wild-type vs. Omicron variant, while, in contrast, the use of non-invasive ventilatory support associated with severe but not critical disease did not appear to mediate much of this effect. The results of this analysis, although underpowered, suggest that severe disease and/or the psycho-physiological stress of ICU admission and procedures are potentially on the causal pathway from infection to the development of PCC [58].

The main limitation of our study is that it cannot be used to obtain a reliable estimate of the incidence of PCC as we discussed before. Other limitations are as follows: (i) the small sample size of the vaccinated participants, which limited the investigation of the role of COVID-19 vaccination as a confounder or as an effect measure modifier of the risk of PCC; of note, the low rate of vaccination was mainly due to the fact that participants acquired the infection when vaccination was not available, as opposed to vaccine hesitancy during the Delta and Omicron phase; (ii) only a small proportion of participants were infected with the newly circulating Omicron strains, and genomics data were available only for a small subset of participants; (iii) most participants were followed up at 2–3 months after the acute phase, thus we do not have data on longer-term outcomes; (iv) we used the CDC definition of PCC, i.e., ongoing symptoms at ≥1 month after the acute disease, but other definitions focusing on symptoms at 3 months after acute infection exist; (v) the study protocol lacks of objective measures of PCC; (vi) this study only included previously hospitalized patients, and therefore the applicability of the results to the general population is limited; and, finally, (vii) our mediation analysis was underpowered and the results are valid only under the assumptions of a counterfactual framework (e.g., correct model specification and no unmeasured confounding).

On the other hand, the fact that the main EuCARE POSTCOVID cohort is nested within the EUCARE hospitalized cohort can also be seen as a strength. Indeed, this protocol design allowed us to construct a weighted pseudo-population to try to control the effects of sampling bias. Of course, we cannot completely rule out collider bias, which might only be attenuated using these weights.

## 5. Conclusions

In conclusion, our data suggest that the reduction in the PCC burden over time may be associated with the Omicron phase of the pandemic, and that the difference in risk seen with the Omicron vs. Wuhan virus appeared to be explained by severity of the disease during the acute phase. However, many questions regarding the development and evolution of PCC in the era of currently circulating variants, following or not following treatment with antiviral drugs and vaccination with ≥1 boosters, need further investigation. The possible effect of the use of antivirals during the acute phase and their effect on viral persistence (which in turn has been associated with the risk of PCC) is currently the subject of a separate analysis within the EuCARE consortium. Further analyses are ongoing to evaluate the effect of vaccination and air pollution on the risk of PCC as well as the rate of symptom resolution over time in those diagnosed with the condition.

## Figures and Tables

**Table 1 viruses-16-01500-t001:** EuCARE Hospitalized cohort and patients’ characteristics according to return for Post COVID-19 follow-up.

Characteristics	Total	Returned to PCC and Entered the POSTCOVID Cohort	Not Returned	*p*-Value
	N = 7699	N = 1317	N = 6382	
Age, years				
Median (IQR)	64 (51, 77)	59 (51, 69)	65 (51, 79)	<0.001
18–39	817 (10.6%)	124 (9.4%)	693 (10.9%)	<0.001
40–49	941 (12.2%)	180 (13.7%)	761 (11.9%)	
50–59	1445 (18.8%)	357 (27.1%)	1088 (17.0%)	
60–69	1543 (20.0%)	345 (26.2%)	1198 (18.8%)	
70–79	1360 (17.7%)	220 (16.7%)	1140 (17.9%)	
80+	1593 (20.7%)	91 (6.9%)	1502 (23.5%)	
Female, n (%)	3358 (43.6%)	566 (43.0%)	2792 (43.7%)	0.607
Nationality, n (%)				<0.001
Italian	2437 (31.7%)	523 (39.7%)	1914 (30.0%)	
Not Italian	3714 (48.2%)	619 (47%)	3095 (48.5%)	
Unknown	1548 (20.1%)	175 (13.3%)	1373 (21.5%)	
Comorbidities, n (%)				
≥1	5006 (65%)	885 (67.2%)	4121 (64.6%)	0.069
Asthma	193 (3.6%)	64 (6.6%)	129 (2.9%)	<0.001
Cancer	562 (10.5%)	45 (4.6%)	517 (11.8%)	<0.001
Cerebrovascular	535 (10.0%)	75 (7.7%)	460 (10.5%)	0.009
Chronic kidney disease	663 (12.4%)	43 (4.4%)	620 (14.1%)	<0.001
Liver disease	173 (3.2%)	16 (1.6%)	157 (3.6%)	0.002
Lung disease	540 (10.1%)	73 (7.5%)	467 (10.6%)	0.003
Diabetes	1473 (27.4%)	235 (24.2%)	1238 (28.2%)	0.011
HIV/AIDS	75 (1.4%)	10 (1.0%)	65 (1.5%)	0.278
Heart disease	1306 (24.3%)	198 (20.3%)	1108 (25.2%)	0.001
Hypertension	3040 (56.6%)	539 (55.4%)	2501 (56.9%)	0.386
Immunodeficiency	115 (2.1%)	21 (2.2%)	94 (2.1%)	0.970
Neurological	460 (8.6%)	32 (3.3%)	428 (9.7%)	<0.001
Obesity	1170 (21.8%)	356 (36.6%)	814 (18.5%)	<0.001
Smoking, n (%)	661 (12.3%)	197 (20.2%)	464 (10.6%)	<0.001
Vaccination, n (%)				<0.001
2+ doses	878 (11.4%)	36 (2.7%)	842 (13.2%)	
Viral variant, n (%)				<0.001
Wild-type strain	3014 (39.1%)	916 (69.6%)	2098 (32.9%)	
Alpha	1281 (16.6%)	189 (14.4%)	1092 (17.1%)	
Delta	981 (12.7%)	117 (8.9%)	864 (13.5%)	
Gamma	121 (1.6%)	52 (3.9%)	69 (1.1%)	
Omicron	2302 (29.9%)	43 (3.3%)	2259 (35.4%)	
Blood tests, median (IQR)				
WBC count (10^9^ cells/L)	6.9 (5.1, 9.5)	6.6 (5.0, 9.1)	6.9 (5.1, 9.6)	0.019
Lymphocyte (10^9^ cell/L)	1.0 (0.7, 1.5)	1.0 (0.7, 1.5)	1.0 (0.7, 1.5)	0.206
Platelets (10^9^ cells/L)	215 (165, 282)	213 (165, 273)	216 (165, 284)	0.425
Hemoglobin (g/dL)	13.8 (12.0, 16.4)	13.9 (12.8, 14.9)	13.7 (11.9, 90.0)	0.359
C-reactive protein (mg/L)	35.5 (8.9, 90.0)	68.0 (29.7, 115.9)	28.0 (6.8, 82.2)	<0.001
Creatinine (mg/L)	0.9 (0.7, 1.6)	0.9 (0.7, 1.0)	1.0 (0.7, 2.2)	<0.001
D-dimer (ng/mL)	490.0 (261.0, 1010)	453.0 (259.0, 943.0)	500.0 (263.5, 1030)	0.097

Categorical data are presented as absolute numbers and percentages, quantitative variables as median and Interquartile Range. Chi-square test or Mann Whitney test was used for comparison between patients who returned for a post COVID-19 visit and patients who didn’t return for follow-up after the acute phase. WBC, White Blood Cells.

**Table 2 viruses-16-01500-t002:** Characteristics of hospitalized patients included in the EuCARE POSTCOVID-19 cohort according to viral variant.

Characteristics	Total (N 1317)	Wild-Type Virus (N 916)	Alpha (N 189)	Delta (N 117)	Gamma (N 52)	Omicron (N 43)	*p*-Values
Age, years, median (IQR)	59 (51, 69)	59 (50, 68)	60 (53, 68)	57 (47, 70)	57 (47, 64)	77 (69, 84)	<0.001
Age groups, n (%): 18–39 40–49 50–59 60–69 70–79 ≥80	124 (9.4%) 180 (13.7%) 357 (27.1%) 345 (26.2%) 220 (16.7%) 91 (6.9%)	90 (9.8%) 123 (13.4%) 256 (27.9%) 244 (26.6%) 151 (16.5%) 52 (5.7%)	14 (7.4%) 21 (11.1%) 59 (31.2%) 53 (28.0%) 32 (16.9%) 10 (5.3%)	14 (12%) 24 (20.5%) 25 (21.4%) 23 (19.7%) 21 (17.9%) 10 (8.5%)	6 (11.5%) 11 (21.2%) 14 (26.9%) 18 (34.6%) 3 (5.8%) 0	0 1 (2.3%) 3 (7%) 7 (16.3%) 13 (30.2%) 19 (44.2%)	<0.001
Female, n (%)	566 (43.0%)	382 (41.7%)	75 (39.7%)	52 (44.4%)	33 (63.5%)	24 (55.8%)	0.010
Nationality, n (%): Italian Not Italian Unknown	523 (39.7%) 619 (47.0%) 175 (13.3%)	338 (36.9%) 488 (53.3%) 90 (9.8%)	122 (64.6%) 36 (19.0%) 31 (16.4%)	28 (23.9%) 41 (35.0%) 48 (41.0%)	0 (0.0%) 48 (92.3%) 4 (7.7%)	35 (81.4%) 6 (14.0%) 2 (4.7%)	<0.001
Comorbidities ≥1, n (%):	885 (67.2%)	617 (67.4%)	113 (59.8%)	73 (62.4%)	45 (86.5%)	37 (86%)	<0.001
Asthma	64 (6.6%)	49 (7.2%)	6 (4.7%)	4 (5.5%)	4 (8.0%)	1 (2.5%)	0.646
Cancer	45 (4.6%)	25 (3.7%)	6 (4.7%)	8 (11.0%)	1 (2.0%)	5 (12.5%)	0.006
Cerebrovascular diseases	75 (7.7%)	22 (3.2%)	15 (11.8%)	30 (41.4%)	3 (6.0%)	5 (12.5%)	<0.001
Chronic kidney diseases	43 (4.4%)	24 (3.5%)	4 (3.1%)	6 (8.2%)	2 (4.0%)	7 (17.5%)	<0.001
Liver diseases	16 (1.6%)	13 (1.9%)	0 (0.0%)	2 (2.7%)	0 (0.0%)	1 (2.5%)	0.411
Lung diseases	73 (7.5%)	49 (7.2%)	10 (7.9%)	8 (11%)	2 (4.0%)	4 (10%)	0.622
Diabetes	235 (24.2%)	173 (25.3%)	31 (24.4%)	6 (8.2%)	17 (34.0%)	8 (20.0%)	0.009
HIV/AIDS	10 (1.0%)	8 (1.2%)	1 (0.8%)	1 (1.4%)	0 (0.0%)	0 (0.0%)	0.874
Heart diseases	198 (20.3%)	160 (23.4%)	20 (15.7%)	11 (15.1%)	4 (8.0%)	3 (7.5%)	0.004
Hypertension	539 (55.4%)	372 (54.5%)	71 (55.9%)	38 (52.1%)	31 (62.0%)	27 (67.5%)	0.429
Immunodeficiencies	21 (2.2%)	19 (2.8%)	1 (0.8%)	0 (0.0%)	1 (2.0%)	0 (0.0%)	0.300
Neurological diseases	32 (3.3%)	12 (1.8%)	4 (3.1%)	12 (16.4%)	0 (0.0%)	4 (10.0%)	<0.001
Obesity, n (%)	356 (36.6%)	276 (40.4%)	26 (20.5%)	18 (24.7%)	31 (62.0%)	5 (12.5%)	<0.001
Smoking, n (%)	197 (20.2%)	150 (22%)	28 (22%)	6 (8.2%)	7 (14.0%)	6 (15.0%)	0.042
Vaccination, n (%): 2+ doses	36 (2.7%)	1 (0.1%)	1 (0.5%)	30 (25.6%)	0 (0.0%)	4 (9.3%)	<0.001
Lymphocyte (10^3^ cells/L)	N 1049 1.0 (0.7, 1.5)	N 804 1.0 (0.7, 1.5)	N 126 1.1 (0.8, 1.4)	N 43 0.9 (0.6, 1.4)	N 38 0.9 (0.6, 1.5)	N 38 1.1 (0.6, 2.2)	0.412
C-reactive protein (mg/L)	N 1051 68 (29.7, 115.9)	N 795 70.3 (33, 123)	N 137 54.8 (23.8, 115.5)	N 45 50.7 (27.8, 90.5)	N 36 90.0 (78.3, 102)	N 38 24.2 (11.2, 66.4)	<0.001

Categorical data are presented as absolute numbers and percentages, quantitative variables as median and Interquartile Range. Chi-square test or Kruskal Wallis test was used for comparison among different viral variants.

**Table 3 viruses-16-01500-t003:** Association between SARS-CoV-2 viral variant and Post COVID-19 condition from fitting a logistic regression model.

Viral Variant (N 1317)	PCC (N 918)		Not PCC (N 399)	*p*-Value
Wild-type strain Alpha Delta Gamma Omicron	641 (69.8%) 139 (15.1%) 78 (8.5%) 44 (4.8%) 16 (1.7%)		275 (68.9%) 50 (12.5%) 39 (9.8%) 8 (2.0%) 27 (6.8%)	<0.001
**Viral Variant**	**OR** **(95% CI)**	** *p* ** **-Value**	**aOR** **(95% CI)**	** *p* ** **-Value**
	Unweighted analysis			
Wild-type strain	1	<0.001	1	<0.001
Alpha	1.19 (0.84, 1.70)		1.77 (1.13, 2.78)	
Delta	0.86 (0.57, 1.29)		0.95 (0.45, 2.01)	
Gamma	2.36 (1.10, 5.08)		2.52 (1.08, 5.88)	
Omicron	0.25 (0.13, 0.48)		0.13 (0.03, 0.66)	
	Analysis weighted for viral variant and demographics			
Wild-type strain	1	<0.001	1	<0.001
Alpha	1.15 (0.99, 1.34)		1.74 (1.41, 2.14)	
Delta	0.87 (0.73, 1.03)		1.32 (0.98, 1.76)	
Gamma	2.30 (1.40, 3.79)		2.77 (1.64, 4.70)	
Omicron	0.25 (0.21, 0.29)		0.31 (0.19, 0.51)	
	Analysis weighted for viral variant, demographics and CRP			
Wild-type strain	1	<0.001	1	<0.001
Alpha	1.37 (1.14, 1.64)		1.10 (0.48, 1.44)	
Delta	1.13 (0.93, 1.38)		1.33 (0.94, 1.87)	
Gamma	2.18 (1.04, 4.54)		1.59 (0.73, 3.46)	
Omicron	0.19 (0.16, 0.21)		0.40 (0.24, 0.64)	

Unadjusted analysis, proportion of patients diagnosed with PCC according to viral variant; logistic regression analysis exploring the association between viral variant and PCC. PCC, Post COVID-19 condition. OR, odds ratio; aOR, adjusted odds ratio; CI, confidence interval; CRP, C reactive protein. Comparison between PCC and not PCC by Chi-square test. Multivariable analysis is adjusted for age, gender, comorbidities and calendar time of infection (using restricted cubic splines). Unweighted analysis; model 1: weighted analysis for demographic characteristics and variant; model 2: weighted analysis for demographic characteristics, variant and CRP.

**Table 4 viruses-16-01500-t004:** Association between SARS-CoV-2 viral variant and PCC clusters from fitting logistic regression analysis in hospitalized patients.

(A) Viral Variant and Brain Fog	OR (95% CI)	*p*	AOR (95% CI)	*p*
	Unweighted analysis			
Wild-type strain	1		1	
Alpha	2.25 (1.63, 3.11)	<0.001	1.69 (1.10, 2.60)	0.016
Delta	1.60 (1.07, 2.40)	0.022	0.85 (0.43, 1.67)	0.634
Gamma	1.87 (1.05, 3.31)	0.033	1.11 (0.56, 2.17)	0.772
Omicron	0.63 (0.29, 1.38)	0.246	0.17 (0.05, 0.63)	0.007
	Analysis weighted for viral variant and demographics			
Wild-type strain	1		1	
Alpha	2.20 (1.91, 2.54)	<0.001	1.90 (1.56, 2.30)	<0.001
Delta	1.51 (1.27, 1.80)	<0.001	1.02 (0.75, 1.39)	0.880
Gamma	1.80 (1.24, 2.62)	0.002	1.19 (0.79, 1.80)	0.405
Omicron	0.55 (0.45, 0.67)	<0.001	0.27 (0.16, 0.45)	<0.001
	Analysis weighted for viral variant, demographics and CRP			
Wild-type strain	1		1	
Alpha	2.36 (2.02, 2.76)	<0.001	2.07 (1.67, 2.57)	<0.001
Delta	2.41 (2.02, 2.87)	<0.001	1.49 (1.06, 2.10)	0.021
Gamma	1.93 (1.14, 3.25)	0.014	1.29 (0.73, 2.27)	0.386
Omicron	0.37 (0.31, 0.45)	<0.001	0.17 (0.10, 0.29)	<0.001
**(B) Viral Variant and Respiratory Sequelae**	**OR (95% CI)**	** *p* **	**AOR (95% CI)**	** *p* **
	Unweighted analysis			
Wild-type strain	1		1	
Alpha	1.10 (0.80, 1.51)	0.560	1.12 (0.73, 1.71)	0.602
Delta	0.65 (0.43, 0.99)	0.045	0.61 (0.31, 1.21)	0.159
Gamma	2.26 (1.28, 3.99)	0.005	1.81 (0.92, 3.55)	0.084
Omicron	0.30 (0.13, 0.68)	0.004	0.19 (0.05, 0.68)	0.011
	Analysis weighted for viral variant and demographics			
Wild-type strain	1		1	
Alpha	1.08 (0.94, 1.24)	0.289	1.28 (1.06, 1.55)	0.010
Delta	0.69 (0.58, 0.82)	<0.001	0.90 (0.67, 1.22)	0.512
Gamma	2.20 (1.52, 3.19)	<0.001	2.27 (1.51, 3.42)	<0.001
Omicron	0.28 (0.23, 0.35)	<0.001	0.34 (0.21, 0.56)	<0.001
	Analysis weighted for viral variant, demographics and CRP			
Wild-type strain	1		1	
Alpha	1.36 (1.17, 1.58)	<0.001	2.29 (1.86, 2.82)	<0.001
Delta	0.58 (0.48, 0.69)	<0.001	1.50 (1.07, 2.09)	0.017
Gamma	3.14 (1.80, 5.48)	<0.001	5.18 (2.84, 9.44)	<0.001
Omicron	0.28 (0.24, 0.32)	<0.001	0.94 (0.57, 1.56)	0.814
**(C) Viral Variant and Chronic Fatigue**	**OR (95% CI)**	** *p* **	**AOR (95% CI)**	** *p* **
	Unweighted analysis			
Wild-type strain	1		1	
Alpha	2.51 (1.80, 3.49)	<0.001	3.01 (1.96, 4.63)	<0.001
Delta	1.64 (1.11, 2.42)	0.013	2.22 (1.16, 4.23)	0.015
Gamma	0.90 (0.51, 1.58)	0.707	0.98 (0.51, 1.90)	0.959
Omicron	0.37 (0.18, 0.76)	0.007	0.50 (0.16, 1.61)	0.246
	Analysis weighted for viral variant and demographics			
Wild-type strain	1		1	
Alpha	2.50 (2.17, 2.88)	<0.001	3.13 (2.60, 3.78)	<0.001
Delta	1.62 (1.38, 1.91)	<0.001	2.37 (1.77, 3.17)	<0.001
Gamma	0.85 (0.59, 1.23)	0.385	0.95 (0.63, 1.42)	0.802
Omicron	0.33 (0.28, 0.40)	<0.001	0.50 (0.35, 0.88)	0.013
	Analysis weighted for viral variant, demographics and CRP			
Wild-type strain	1		1	
Alpha	3.04 (2.58, 3.57)	<0.001	4.36 (3.52, 5.41)	<0.001
Delta	2.53 (2.11, 3.03)	<0.001	4.51 (3.25, 6.26)	<0.001
Gamma	0.91 (0.54, 1.53)	0.726	1.22 (0.70, 2.13)	0.481
Omicron	0.25 (0.21, 0.29)	<0.001	0.60 (0.37, 0.99)	0.045

OR, odds ratio; AOR, adjusted odds ratio; CI, confidence interval; CRP, C reactive protein. Multivariable analysis is adjusted for age, gender, comorbidities and calendar time of infection. Unweighted analysis; model 1: weighted analysis for demographic characteristics and variant; model 2: weighted analysis for demographic characteristics, variant and CRP. (A) Outcome: brain fog; (B) Outcome: respiratory sequelae; (C) Outcome: fatigue.

**Table 5 viruses-16-01500-t005:** Four-way decomposition mediation analysis for the effect of wild-type strain versus Omicron variant on PCC and brain fog, admission in ICU/MV and CPAP/NIV as mediators.

PCC	Wild-Type vs. Omicron Four-Way Decomposition—Binary Outcome PCC, Admission in MV-ICU as the Mediator	
Component	Excess RR (95% CI)	Proportion Attributable (95% CI), % *p*-value
CDE ^1^	3.41 (−1.53, 8.34)	74.8 (53.6, 95.9)
		<0.001
INT-ref ^2^	−0.75 (−2.18, 0.67)	−16.5 (−36.2, 3.2)
		0.100
INT-med ^3^	1.65 (−0.79, 4.10)	36.3 (15.0, 57.6)
		<0.001
PIE ^4^	0.25 (−0.05, 0.55)	5.4 (−3.4, 14.3)
		0.228
TERR ^5^	4.56 (−1.48, 10.60)	100.0
ORTE ^6^	5.56 (1.00, 11.60)	
Overall proportion due to Interaction		19.8 (5.1, 34.4)
Overall proportion due to Mediation		41.7 (14.0, 69.5)
**PCC**	**Wild-Type vs. Omicron Four-Way Decomposition—Binary Outcome PCC,** **NIV-CPAP as the Mediator**	
Component	Excess RR (95% CI)	Proportion Attributable (95% CI), % *p*-value
CDE ^1^	3.81 (0.20, 7.42)	107.3 (88.3, 126.4)
		<0.001
INT-ref ^2^	−0.09 (−0.67, 0.49)	−2.6 (−20.9, 15.6)
		0.779
INT-med ^3^	−0.04 (−0.30, 0.22)	−1.0 (−7.6, 5.6)
		0.766
PIE ^4^	−0.13 (−0.36, 0.10)	−3.7 (−12.7, 5.3)
		0.418
TERR ^5^	3.55 (−0.31, 7.41)	100.0
ORTE ^6^	4.55 (1.00, 8.41)	
Overall proportion due to Interaction		0.0 (0.0, 8.5)
Overall proportion due to Mediation		0.0 (0.0, 4.5)
**Brain Fog**	**Wild-Type vs. Omicron Four-Way Decomposition—Binary Outcome Brain Fog,** **Admission in MV-ICU as the Mediator**	
Component	Excess RR (95% CI)	Proportion Attributable (95% CI), % *p*-value
CDE ^1^	−0.41 (−1.08, 0.27)	108.3 (59.7, 157.0)
		<0.001
INT-ref ^2^	−0.00 (−0.08, 0.08)	1.0 (−20.8, 22.8)
		0.930
INT-med ^3^	0.01 (−0.08, 0.09)	−1.3 (−24.3, 21.6)
		0.908
PIE ^4^	0.03 (−0.14, 0.20)	−7.9 (−56.4, 40.5)
		0.748
TERR ^5^	−0.37 (−1.03, 0.28)	100.0
ORTE ^6^	0.63 (1.00, 1.28)	
Overall proportion due to Interaction		−0.4 (0.0, 1.3)
Overall proportion due to Mediation		−9.3 (0.0, 28.5)
**Brain Fog**	**Wild-Type vs. Omicron Four-Way Decomposition—Binary Outcome Brain Fog, NIV-CPAP as the Mediator**	
Component	Excess RR (95% CI)	Proportion Attributable (95% CI), % *p*-value
CDE ^1^	1.14 (−0.29, 2.58)	228.9 (−261, 719.3)
		0.360
INT-ref ^2^	−0.69 (−1.23, −0.15)	−138 (−652, 376.6)
		0.600
INT-med ^3^	0.27 (−0.14, 0.67)	53.1 (−166, 272.1)
		0.635
PIE ^4^	−0.22 (−0.58, 0.14)	−44.4 (−237, 147.9)
		0.651
TERR ^5^	0.50 (−1.14, 2.14)	100.0
ORTE ^6^	1.50 (1.00, 3.14)	
Overall proportion due to Interaction		0.0 (0.0, 100.0)
Overall proportion due to Mediation		8.7 (0.0, 37.5)

4-way decomposition method for the mediation analysis to estimate what proportion of the total effect associated with the wild-type virus (using the Omicron variant as a comparator) on PCC might be mediated by disease severity. The need for mechanical ventilation (MV) and Intensive Care Unit (ICU) admission vs. no oxygen therapy and the use of Continuous Positive Airway Pressure (CPAP), or Non-Invasive Mechanical Ventilation (NIV) vs. no oxygen treatment (in 2 separate models) was used to classify participants with severe and moderate disease and used as a mediator for the difference in PCC risk between the Wuhan and Omicron strains. We analysed the effect of wild-type strain vs. Omicron variant on PCC and brain fog. ^1^ Controlled Direct Effect (neither mediation not interaction). ^2^ Reference Interaction (interaction but not mediation). ^3^ Mediated Interaction (both mediation and interaction). ^4^ Pure Indirect Effect (mediation but no interaction). ^5^ Total Excess Relative Risk. ^6.^ Odds Ratio Total Effect. The model was adjusted for age, sex and comorbidities.

## Data Availability

The raw data supporting the conclusions of this article will be made available by the authors upon request.

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
