# Peer review of "The Omicron Variant Is Associated with a Reduced Risk of the Post COVID-19 Condition and Its Main Phenotypes Compared to the Wild-Type Virus: Results from the EuCARE-POSTCOVID-19 Study"

_viruses, 2024, doi:10.3390/v16091500_

Round 1
Reviewer 1 Report
Comments and Suggestions for Authors
This manuscript presents important analysis to assess the risk of Post-Covid condition (PCC) among patients infected with different strains of SARS-CoV2 who participated in a multicenter cohort study as ap part of EuCARE project. Authors investigated whether viral variants are associated with the development of symptoms such as fatigue, brain fog and respiratory problems and only well selected group of previously hospitalized patients was included in this study. Bai et al. applied regression modelling inverse probability weighting to minimize biases introduced by potential confounders. Interestingly, authors classified patients with PCC using the clusters of similar symptoms proposed for the other post-acute infection syndrome, MECFS.
In summary, this analysis provides valuable insights into the long-term implications of different SARS-CoV2 strains on patients outcomes. However, there are some comments to be addressed:
1. Abstract section: a) line 35- the word “viral” that describes variants is missing; b) line 36- add number of patients included in the study; c) line 38- the abbreviation “GISAID” is used in the manuscript for the first time, what does it stand for ?; d) line 43-include percentage of alpha, delta and gamma strains in in all evaluated patients; e) line 45-there are data missing related to evaluated non-PCC patients; f) line 48- the word “non-negligible” to be replaced with the word “significant”.
2. Introduction section: a) line 85- indicate when 2 strains: delta and omicron were contracted (similar to 3 other strains mentioned in the same paragraph); b) line 102- add word “viral” to the word “variants”; line 104- delete the word “syndrome”.
3. Material and methods section: a) line 111 and 113- the frequency of hospitalized and evaluated patients; b) Figure 1 to be moved to supplemental material; c) line 129- enlist the blood tests performed after the acute Covid infection; d) line 130- add reference to post Covid-19 WHO Case Report Form; e) line 145- describe how viral sequencing was done; f) line 151- a new term “variants of concern” may be confusing for some readers and therefore should be removed from this manuscript; g) line 170- delete “SARS Cov2”; f) line 191- add reference; h) line 198-there is no information referring to the supplemental Figure 1; i) line 206- what does “CRP” stand for?; j) line 209- remove “VoC”; k) line 212- explain why Omicron variant was used as comparator.
4. Results section (this is very confusing part of the manuscript, it’s difficult to follow and it needs to be re-written); a) subsection 3.1: line 224-add Supplemental Table 2 to the main text (as a new Table 1), describe it in this subsection and remove all the information regarding previous Table 1; line 234 and 235- add median age and frequency of male participants and indicate what nationalities were included in the non-Italian study population; show the race of all patients enrolled in the study (n=7,699); b) subsection 3.2: describe only new Table 2 (previous Table 1) including vaccination and lymphocyte frequency data and present it on one page instead of dividing it into 2 parts; c) subsection 3.3: to be removed (current subsection 3.3 will be included in the new subsection 3.2); d) subsection 3.4 and 3.5-only refers to the new Table 3 (previous Table 2) and Supplementary Table 3; e) subsection 3.6 and 3.7- refers to the new Table 4 and 5 (previous Table 3 and 4), respectively.
5. Discussion section: a) how different are these results from the previously published report (e.g. reference #23); b) all the limitations authors described in the second paragraph (line 337) should be moved to the end of this section (line 396).
Comments on the Quality of English Language
to be improved
Author Response
Open Review
(x) I would not like to sign my review report
( ) I would like to sign my review report
Quality of English Language
( ) I am not qualified to assess the quality of English in this paper.
( ) The English is very difficult to understand/incomprehensible.
( ) Extensive editing of English language required.
(x) Moderate editing of English language required.
( ) Minor editing of English language required.
( ) English language fine. No issues detected.
This manuscript presents important analysis to assess the risk of Post-Covid condition (PCC) among patients infected with different strains of SARS-CoV2 who participated in a multicenter cohort study as ap part of EuCARE project. Authors investigated whether viral variants are associated with the development of symptoms such as fatigue, brain fog and respiratory problems and only well selected group of previously hospitalized patients was included in this study. Bai et al. applied regression modelling inverse probability weighting to minimize biases introduced by potential confounders. Interestingly, authors classified patients with PCC using the clusters of similar symptoms proposed for the other post-acute infection syndrome, MECFS.
In summary, this analysis provides valuable insights into the long-term implications of different SARS-CoV2 strains on patients outcomes. However, there are some comments to be addressed:
We thank the reviewer for the general positive assessment and time and effort devoted to improve our manuscript.
- Abstract section:
- a) line 35- the word “viral” that describes variants is missing;
We have now added the word “viral” (page 2, line 2).
- b) line 36- add number of patients included in the study;
We have now added the number of the enrolled patients in the EuCARE hospitalized cohort (page 2, line 2).
- c) line 38- the abbreviation “GISAID” is used in the manuscript for the first time, what does it stand for ?;
GISAID is a global data science initiative and the primary source of genomic and associated metadata of all influenza viruses, Respiratory Syncytial Virus (RSV) and severe acute respiratory syndrome coronavirus 2 (SARS-CoV-2), available at https://gisaid.org.
Khare, S., et al (2021) GISAID’s Role in Pandemic Response. China CDC Weekly, 3(49): 1049-1051.doi: 10.46234/ccdcw2021.255 PMCID: 8668406
Elbe, S. and Buckland-Merrett, G. (2017) Data, disease and diplomacy: GISAID’s innovative contribution to global health. Global Challenges, 1:33-46.
doi: 10.1002/gch2.1018 PMCID: 31565258
Shu, Y. and McCauley, J. (2017) GISAID: from vision to reality. EuroSurveillance, 22(13) doi: 10.2807/1560-7917.ES.2017.22.13.30494 PMCID: PMC5388101
We have now spelled out the acronym in the Abstract (page 2, line 4) and added the relevant references.
- d) line 43-include percentage ofalpha, delta and gamma strains in in all evaluated patients;
We have now added in the text of the Abstract the breakdown of the viral variants in the study population of 1,317 individuals included for analysis. We have also reported the proportions of participants carrying specific variants among the whole of the EuCARE HOSPITALIZED (page 2, lines 8-9; page 7, lines 205-207) and POSTCOVID cohorts (page 8, lines 226-228; Table 2).
- e) line 45-there are data missing related to evaluated non-PCC patients;
“Omicron vs WT was associated with a reduced risk of PCC and PCC-clusters; conversely, we observed a higher risk with Delta and Alpha variant vs WT.”
We are unclear to which missing data the reviewer is referring to. We are happy to modify the sentence upon further clarification of this request.
- f) line 48- the word “non-negligible” to be replaced with the word “significant”.
We respectfully disagree with this suggestion. We do not think that “significant” is the right word in this context as it may be confused with “statistical significance”.
- Introduction section:
- a) line 85- indicate when 2 strains: delta and omicron were contracted (similar to 3 other strains mentioned in the same paragraph);
We have now reported the calendar periods in which Omicron and Delta variants infections had occurred according to the cited manuscript (page 3, lines 78-79).
- b) line 102- add word “viral” to the word “variants”; line 104- delete the word “syndrome”.
We have now added the word “viral” and we have now deleted the word “syndrome” as suggested.
- Material and methods section:
- a) line 111 and 113- the frequency of hospitalized and evaluated patients;
We respectfully disagree with this suggestion. The number of participants in these cohorts are part of the results so they should not be reported in the Methods section.
- b) Figure 1 to be moved to supplemental material;
We have now moved Figure 1 in the supplemental material (supplementary Figure 1) as suggested.
- c) line 129- enlist the blood tests performed after the acute Covid infection;
We have now added the list of blood tests that were conducted in participants at each follow-up visit (page 5, lines 133-134).
- d) line 130- add reference to post Covid-19 WHO Case Report Form;
We have now added the requested reference (page 5, lines 135).
Reference number 33: World Health Organization, W. Global COVID-19 Clinical Platform Case Report Form (CRF) for Post COVID condition (Post COVID-19 CRF). Updated on 25 February 2021.
- e) line 145- describe how viral sequencing was done;
The whole genome has been sequenced, often including multiple genomic regions or only the spike gene according to the methodology adopted in the different participating centers during the study period. We have added this information in the revised Methods section (page 5, lines 147-149).
- f) line 151- a new term “variants of concern” may be confusing for some readers and therefore should be removed from this manuscript;
We agree with this, especially for the ancestor Wuhan strain. We have now replaced the term “variants of concern” with “viral variants” throughout the manuscript.
- g) line 170- delete “SARS Cov2”;
We have now deleted SARS CoV-2.
- f) line 191- add reference;
We have now added the relevant reference (page 6, line 171).
- h) line 198-there is no information referring to the supplemental Figure 1;
We thank the reviewer for spotting this oversight.
We have used a directed acyclic graph (DAG) to illustrate why our estimates might be affected by collider bias. A collider is indeed defined as a variable that is the consequence of two other variables, for example in our case when our exposure of interest (the viral variant) and the outcome (risk of PCC) both affect the likelihood of being sampled (they “collide” in the depicted DAG). Colliders is an issue here because by restricting the analysis only to the sample of hospitalised patients who returned for a PCC visit we are essentially conditioning on a collider and therefore potentially introducing a spurious association between exposure and outcome.
We have added a footnote explaining the rationale for including the Figure.
- line 206- what does “CRP” stand for?;
We have now spelled out the acronym CRP (C reactive protein).
- j) line 209- remove “VoC”;
We have now replaced the word “VoC” with “viral variant” (see our response to a previous point).
- k) line 212- explain why Omicron variant was used as comparator.
The biggest difference in risk of PCC was observed between the Omicron and Wuhan strains. Because most software performing mediation analysis using a counterfactual framework can only handle a binary exposure we used the contrast Wuhan vs. Omicron for the mediation analysis. The only reason for choosing Omicron as the comparator (instead of Wuhan) was to provide odds ratios >1 (increase in risk) which are generally easier to interpret.
- Results section (this is very confusing part of the manuscript, it’s difficult to follow and it needs to be re-written);
We have now tried to rearrange this section to improve readability.
- subsection 3.1: line 224-add Supplemental Table 2 to the main text (as a new Table 1), describe it in this subsection and remove all the information regarding previous Table 1; line 234 and 235- add median age and frequency of male participants and indicate what nationalities were included in the non-Italian study population; show the race of all patients enrolled in the study (n=7,699);
We have now made these changes as suggested. The table below shows the breakdown of ethnicity according to participants nationality. As expected, the majority of Italians were of Caucasian ethnicity while a high proportion of the foreigners were Hispanic (participants enrolled in Lisbon or Brazil sites). Approximately 15% of participants had missing value for ethnicity and this is why we preferred not to use this factor.
|
Ethnicity |
Italian |
Non-Italian |
Total |
|
|
|
|
|
|
African |
1 (0.1) |
9 (1.5) |
10 (0.8) |
|
Arabic |
0 (0) |
25 (4.0) |
25 (1.9) |
|
Asian |
2 (0.3) |
18 (2.9) |
20 (1.5) |
|
Caucasian |
565 (81.0) |
86 (13.9) |
651 (49.4) |
|
Hispanic |
2 (0.3) |
417 (67.4) |
419 (31.8) |
|
Other/Unknown |
128 (18.3) |
64 (10.3) |
192 (14.6) |
|
|
|
|
|
|
Total |
698 (100.0) |
619 (100.0) |
1,371 (100.0) |
- subsection 3.2: describe only new Table 2 (previous Table 1) including vaccination and lymphocyte frequency data and present it on one page instead of dividing it into 2 parts;
We have now made these changes as suggested.
- c) subsection 3.3: to be removed (current subsection 3.3 will be included in the new subsection 3.2);
We have now made these changes as suggested.
d)subsection 3.4 and 3.5-only refers to the new Table 3 (previous Table 2) and Supplementary Table 3;
We have now made these changes as suggested.
- subsection 3.6 and 3.7- refers to the new Table 4 and 5 (previous Table 3 and 4), respectively.
We have now made these changes as suggested.
- Discussion section:
- a) how different are these results from the previously published report (e.g. reference #23);
We have now revised the Discussion section to incorporate a more detailed comparison between the results of our analysis and those of previously published studies.
- b) all the limitations authors described in the second paragraph (line 337) should be moved to the end of this section (line 396).
We have now moved all the limitations at the end of the section as suggested.
Reviewer 2 Report
Comments and Suggestions for Authors
This study investigated the relationship between different COVID-19 variants and the development of long-term symptoms (Post-COVID Condition or PCC). Researchers analyzed data from hospitalized patients across six centers between 2020 and 2023. Key findings were: Omicron variant was associated with a lower risk of PCC compared to the original wild-type virus; Delta and Alpha variants were linked to a higher risk of PCC compared to the wild-type. The most common PCC symptoms were fatigue, brain fog, and respiratory issues. A significant portion of the increased PCC risk associated with the wild-type virus compared to Omicron was attributed to the higher likelihood of ICU admission for patients infected with the wild-type strain. Overall, the study suggests that the risk of developing long-term COVID symptoms has decreased over time with the emergence of newer variants, but the severity of the initial illness continues to be a significant factor in PCC development. The manuscript presents an interesting analysis of the relationship between COVID-19 variants and the risk of Post-COVID Condition (PCC). However, several areas require clarification and improvement to enhance the clarity and robustness of the study.
Major comments:
- Clearly state the primary and secondary objectives of the study. What is the specific hypothesis being tested?
- Outline the anticipated findings or results based on the study objectives.
- Cohort division is inconsistent. While dividing the cohort based on vaccination status, infection type, age, and comorbidities is a valuable approach, consider the potential for subgroup analysis rather than creating separate cohorts. This allows for a more comprehensive analysis of the interaction between these variables and PCC risk.
- Patient assessment and follow-up is not discussed. Provide detailed information about the assessment and follow-up protocol for acute COVID-19 patients, including data collection methods for symptoms, comorbidities, and other relevant variables.
- How were symptoms recorded over time for participants to assess PCC? Were there any missing data for key variables, and how were these handled?
- Strengthen the discussion section by providing a more in-depth interpretation of the results considering dividing into smaller cohorts.
Minor comments:
1. Vaccinated and unvaccinated patients: Clearly justify the decision to include both vaccinated and unvaccinated patients in the same cohort or separate cohorts. If combined, provide a rationale for this approach.
2. Secondary cluster analysis: Explain how vaccination status was considered in the secondary cluster analysis. It is crucial to assess the impact of vaccination on PCC clusters. It would be nice to keep them separate.
3. Infection types: Clarify if the analysis included only patients with a single infection (WT, Alpha, Delta, or Omicron) or if mixed infections (like WT encounters followed by Omicron) were considered. If mixed infections were excluded, please ignore. Since it is highly likely to have infected with prevalent Omicron variants after first encounter with WT or delta and so on.
4. Age groups: Compare the cohorts for age differences and assess the impact of age on PCC risk and symptom severity for example old versus young.
5. Omicron strain: Specify the Omicron variant(s) included in the analysis.
6. Figure and table quality: Improve the quality of Figure 1 and Table 3 for better clarity and readability. Consider using a flowchart for Figure 1 and restructuring Table 3 for better organization.
Author Response
Major comments:
- Clearly state the primary and secondary objectives of the study. What is the specific hypothesis being tested?
We thank the reviewer for asking this clarification. Previous data showed a trend of reduction of post COVID-19 condition (PCC) when comparing Omicron with Alpha/Delta or wild-type waves. Also a change in the phenotypes of PCC (a reduction in anosmia/dysgeusia and cardiac or respiratory symptoms, while a stability in the incidence of fatigue, brain fog and pain) has been documented.
Given these premises, the main hypothesis of our study was that the risk of PCC might be reduced following infection with recent viral variants compared to that seen in early waves of the pandemic. Having experienced a lower severity of COVID-19 disease during the acute phase is one of the possible mechanisms that could explain this reduction in risk over time, which we have also investigated.
Thus, the primary objective of our analysis was to evaluate the association between SARS CoV-2 variants at time of primary SARS CoV-2 infection and risk of developing post COVID-19 condition after recovering from the acute infection.
Secondary aims were i) to investigate whether the predictive role of variants might be different according to specific post COVID-19 condition phenotypes/clusters (i.e. brain fog, respiratory symptoms and fatigue) and ii) to estimate the proportion of total effect of variants on risk of developing PCC that could be explained by disease severity during the acute phase.
We have now added the hypothesis, primary and secondary objectives of the study in the Introduction and Methods sections of the manuscript (page 5, lines 121-130).
We have also updated our literature research and added some new References in the Introduction section.
- Outline the anticipated findings or results based on the study objectives.
We have now highlighted the key findings of our analysis based on the study objectives at the beginning of the Discussion section (pages 9-10, lines 278-286).
These are summarized as outlined below:
- The Omicron variant was associated with a reduction in the risk of PCC, while Alpha and Delta variants were characterized by a higher risk of PCC, compared to the wild-type virus.
- The direction and magnitude of the associations seen for variants and the CDC PCC outcome were similar when we evaluated the risk of specific PCC clusters (brain fog, respiratory symptoms and fatigue).
- An important proportion of the total effect of Omicron vs. wild type variant on the risk of PCC appeared to be explained by disease severity in the acute phase.
- Cohort division is inconsistent. While dividing the cohort based on vaccination status, infection type, age, and comorbidities is a valuable approach, consider the potential for subgroup analysis rather than creating separate cohorts. This allows for a more comprehensive analysis of the interaction between these variables and PCC risk.
We did not perform stratified analyses by age and comorbidities. The potential confounding effect of these variables were controlled for by regression adjustment. We agree that vaccination is likely to be both a confounder and an effect measure modifier for the association of interest and we did intend to perform a formal statistical test for interaction. Nevertheless, because of the very small proportion of vaccinated participants (n=36, 3%), the power of the interaction test would be too low. For this reason, we decided to control for vaccination by conducting a sensitivity analysis after excluding the few vaccinated individuals.
- Patient assessment and follow-up is not discussed. Provide detailed information about the assessment and follow-up protocol for acute COVID-19 patients, including data collection methods for symptoms, comorbidities, and other relevant variables.
All patients included had been hospitalized during their acute SARS CoV-2 infection; after clinical recovery and discharge, they were followed-up in a post COVID clinic at each of the participating centers. According to the original EuCARE POSTCOVID study protocol, patients enrolled in the PCC cohort are scheduled to be seen at 2-3 months (T0, entry in PCC cohort), 6-9 months (T1) and 12-15 months (T2) after the acute phase; the visits could be performed either in person or by telemedicine.
In this analysis we included individuals with ≥1 follow-up evaluation over the first 15 months after the acute phase. Most PCC diagnoses (854/918, 93%) occurred at T0. Median time from the date of hospital discharge to T0 was 2.6 months (IQR 1.84-3.97).
We have referenced in the Methods section the following paper: Varisco, B.; Bai, F.; De Benedittis, S.; Tavelli, A.; Cozzi-Lepri, A.; Sala, M.; Miraglia, F.G.; Santoro, M.M.; Ceccherini-Silberstein, F.; Shimoni, Y.; et al. EuCARE-POSTCOVID Study: a multicentre cohort study on long-term post-COVID-19 manifestations. BMC Infect Dis 2023, 23, 684, doi:10.1186/s12879-023-08595-0. which provides further details of the study protocol and participants management (page 4, lines 103-112).
Regarding the collection of symptoms related to PCC, at each follow-up visit, participants underwent blood tests, were visited by a physician and filled in a short version of the WHO CRF for long COVID (see the following questionnaire; we have now added the reference: “World Health Organization, WHO, Global COVID-19 Clinical Platform Case Report Form (CRF) for Post COVID condition (Post COVID-19 CRF); Updated on 25 February 2021; available at https://www.who.int/publications/i/item/global-covid-19-clinical-platform-case-report-form-(crf)-for-post-covid-conditions-(post-covid-19-crf-) see also image inserted below).
We have now added a more detailed description of the clinical follow-up and data collection in the Methods section of this revised version of the paper (page 5, lines 132-137).
- How were symptoms recorded over time for participants to assess PCC? Were there any missing data for key variables, and how were these handled?
Symptoms were investigated during the visit at the post COVID Clinic (either in person or by telemedicine) using the short version of the post COVID-19 WHO Case Report Form (see above). Typically, if a participant filled in a WHO CRF, he/she would complete all the key variables for PCC assessment. However, at the time of this analysis, not many participants have been followed-up past the entry visit, so the analysis is essentially cross-sectional at time T0. We have further clarified this aspect in the Methods.
- Strengthen the discussion section by providing a more in-depth interpretation of the results considering dividing into smaller cohorts.
We have now rephrased the Discussion section to provide a more clear and in-depth interpretation of the results. We did not set up specific hypotheses on possible effect measure modifiers, so we do not see the scientific rationale or benefit for presenting stratified analyses.
Minor comments:
- Vaccinated and unvaccinated patients: Clearly justify the decision to include both vaccinated and unvaccinated patients in the same cohort or separate cohorts. If combined, provide a rationale for this approach.
The main analysis has been performed on the combined cohort including both vaccinated and unvaccinated patients. However, because vaccinated individuals have a lower risk of severe disease and possibly consequently a lower risk of developing PCC in the long-term, we have also performed a sensitivity analysis after excluding the vaccinated population. Unfortunately, because the sample size of the vaccinated participants was very small (n=36) we could not show the results in the vaccinated stratum.
- Secondary cluster analysis: Explain how vaccination status was considered in the secondary cluster analysis. It is crucial to assess the impact of vaccination on PCC clusters. It would be nice to keep them separate.
We agree with the reviewer. Indeed, the sensitivity analyses after excluding the vaccinated population has been conducted symmetrically for the main PCC outcome and for all secondary clusters outcomes. Results have been included in Supplementary materials (Supplementary Table 2 and 3, page 8 lines 250-253; page 9, lines 269-271).
- Infection types: Clarify if the analysis included only patients with a single infection (WT, Alpha, Delta, or Omicron) or if mixed infections (like WT encounters followed by Omicron) were considered. If mixed infections were excluded, please ignore. Since it is highly likely to have infected with prevalent Omicron variants after first encounter with WT or delta and so on.
We thank the reviewer for asking this clarification.
Participants for whom sequencing was available (n=37, 2%), the sequence did not show evidence for mixed infections or reinfections.
For the remaining participants, the variant was estimated using publicly available data and therefore the single most frequently circulating variant has been assigned to them.
- Age groups: Compare the cohorts for age differences and assess the impact of age on PCC risk and symptom severity for example old versus young.
We did not analyse the impact of age on risk of developing PCC because it was beyond the scope of our analysis which solely focused on variants as the exposure of interest; however, age was treated as a potential confounding factor in all analyses. The table below shows the univariable association between age and risk of PCC indicating that was a weak predictor of the outcome in our dataset.
Table. Association between age at hospital admission and risk of PCC
|
Age group, years |
PCC n (%) |
Total |
Chi-square test p-value |
|
|
|
|
0.25 |
|
18-39 |
79 (64%) |
124 |
|
|
40-59 |
127 (71%) |
180 |
|
|
50-59 |
244 (68%) |
357 |
|
|
60-69 |
247 (72%) |
345 |
|
|
70-79 |
163 (74%) |
220 |
|
|
80+ |
58 (64%) |
91 |
|
|
|
|
|
|
|
Total |
|
1,317 |
|
- Omicron strain: Specify the Omicron variant(s) included in the analysis.
The table below shows the breakdown of the 43 participants infected with Omicron variants by sub lineage (using both sequencing and estimated data). We have added this information in the text of the revised Results section (page 8, lines 227-228).
|
Omicron sub lineage |
N (%) |
|
Omicron (B.1.1.529 BA.1) |
8 (19%) |
|
Omicron (B.1.1.529 BA.2) |
15 (35%) |
|
Omicron (B.1.1.529 BA.5) |
15 (35%) |
|
Omicron (BQ.1.1) |
5 (11%) |
|
|
|
|
Total |
43 (100%) |
- Figure and table quality: Improve the quality of Figure 1 and Table 3 for better clarity and readability. Consider using a flowchart for Figure 1 and restructuring Table 3 for better organization.
We agree with the reviewer, and we have now included a flow-chart as Figure 1 and have restructured Table 3 in order to improve its clarity (reduction of number of lines and improvement in readability).
Reviewer 3 Report
Comments and Suggestions for Authors
The study is retrospective and prospective multicenter cohort study aiming to assess the risk of Post COVID-19 condition among previously hospitalized patients, according to the original SARS-CoV-2 strain and subsequent variants. The work presented for review concerns an important topic. The research is interesting and presented correctly. I have some comments as below.
Minor revisions:
- In the Results section (3.2. Demographic and clinical characteristic of participants according to the inclusion in the EuCARE-POSTCOVID cohort) author notes: “To assess the potential impact of collider bias on our analysis, we compared hospitalized patients who attended ≥1 follow-up visit with those who never returned for evaluation: the former group tended to be younger and was generally less likely to have ≥1 comorbidity”. However, in Supplementary Table 2, the percentage of patients with ≥1 comorbidities was 67.2% in the group Returned to PCC compared to 64.6% in the group Not returned, and the difference was not significant.
- The unit of measurement for lymphocytes in the Supplementary Table 2 is incorrect. It should be 109cell/L instead of 103cell/L. Correct, please.
- In the Results section (3.3. Participants’ characteristics according to SARS CoV-2 viral variant) author notes that the hypertension was more common with the wild-type variant among participants entering the post COVID-19 cohort. However, in the Table 1, hypertension is more common in patients with the Omicron variant, and the difference was not significant.
- In Table 1, the row “Female, n (%)” is repeated. Correct, please.
- Are the differences presented in the study significant after adjusting for multiple comparisons?
Author Response
The study is retrospective and prospective multicenter cohort study aiming to assess the risk of Post COVID-19 condition among previously hospitalized patients, according to the original SARS-CoV-2 strain and subsequent variants. The work presented for review concerns an important topic. The research is interesting and presented correctly. I have some comments as below.
We thank the reviewer for the general positive assessment and time and effort devoted to improve our manuscript.
Minor revisions:
- In the Results section (3.2. Demographic and clinical characteristic of participants according to the inclusion in the EuCARE-POSTCOVID cohort) author notes: “To assess the potential impact of collider bias on our analysis, we compared hospitalized patients who attended 1 follow-up visit with those who never returned for evaluation: the former group tended to be younger and was generally less likely to have ≥1 comorbidity”. However, in Supplementary Table 2, the percentage of patients with ≥1 comorbidities was 67.2% in the group Returned to PCC compared to 64.6% in the group Not returned, and the difference was not significant.
We thank the reviewer for spotting this inconsistency. This was an oversight from our side. We have now reworded the specific sentence in the revised section 3.2 as follows:
“To assess the potential impact of collider bias on our analysis, we compared hospitalized patients who attended 1 follow-up visit with those who never returned for evaluation: the former group tended to be younger and with a lower prevalence of preexisting comorbidities, although asthma, obesity and smoking were more common in PCC cohort participants compared to patients who never returned for a PCC visit.”
- The unit of measurement for lymphocytes in the Supplementary Table 2 is incorrect. It should be 109cell/L instead of 103cell/L. Correct, please.
We thank the reviewer for spotting this typo; we have now corrected the unit of measurement in Supplementary Table 2.
- In the Results section (3.3. Participants’ characteristics according to SARS CoV-2 viral variant) author notes that the hypertension was more common with the wild-type variant among participants entering the post COVID-19 cohort. However, in the Table 1, hypertension is more common in patients with the Omicron variant, and the difference was not significant.
We thank the reviewer for spotting this inconsistency. We have now rearranged the whole of the Result section, so previous paragraph 3.3 has been now merged into section 3.2. We have also rephrased the specific sentence as follows:
“Participants infected by the Omicron variant were generally older, Italian and more likely to have at least one pre-existing comorbidity, with the exception of diabetes, heart diseases, smoking and obesity that were more common in participants infected by the wild-type strain.”
- In Table 1, the row “Female, n (%)” is repeated. Correct, please.
We than the reviewer for spotting this repetition. The description of sex at birth is now included in the new Table 2 (previously Table 1).
- Are the differences presented in the study significant after adjusting for multiple comparisons?
There was no formal correction for multiple comparisons as all analyses have been planned at the outset. Nevertheless, we did repeat the same model for 4 possible outcome (1 primary outcome and 3 additional secondary clusters outcomes) and therefore may be advisable to make a correction to account for possible inflation of the type I error.
However, even using the conservative significance level of 0.0125 (Bonferroni adjusted p-value 0.05/4=0.0125) the data still carry evidence that, compared to participants infected with the Wuhan strain, those infected with Omicron were at lower risk while those infected with Alfa/Delta at higher risk of developing PCC. We have now added a sentence in the Results to note that associations would be significant even at the Bonferroni-corrected significance level of 1.25% (page 9, lines 266-269).
Round 2
Reviewer 1 Report
Comments and Suggestions for Authors
Authors addressed the comments sent in the initial review process.
Comments on the Quality of English LanguageMinor revision on the quality of English is suggested.